# LEARNING INVARIANCES FOR POLICY GENERALIZATION

**Remi Tachet des Combes, Philip Bachman & Harm van Seijen**
Microsoft Research Montreal
Montreal, QC H3A 3H3, CANADA
`{retachet,phbachma,havansei}@microsoft.com`

## ABSTRACT

While recent progress has spawned very powerful machine learning systems, those agents remain extremely specialized and fail to transfer the knowledge they gain to similar yet unseen tasks. In this paper, we study a simple reinforcement learning problem and focus on learning policies that encode the proper invariances for generalization to different settings. We evaluate three potential methods for policy generalization: data augmentation, meta-learning and adversarial training. We find our data augmentation method to be effective, and study the potential of meta-learning and adversarial learning as alternative task-agnostic approaches.

## 1 INTRODUCTION

Deep Reinforcement Learning has produced impressive results in the recent past, allowing machines to master Go (Silver et al., 2017), beat Ms. PacMan (van Seijen et al., 2017) or achieve complex robotic locomotion (Schulman et al., 2015). The algorithms used to achieve those feats are fairly generic, but their outcome is extremely specialized and cannot be used to play any other game: the methods developed for AlphaGo can be used on chess or shogi, but AlphaGo itself cannot play those games. The knowledge those systems gain does not transfer to other scenarios. Designing training algorithms and artificial agents that perform well out-of-the-box on tasks not encountered before is referred to as *zero-shot learning* (Oh et al., 2017; Higgins et al., 2017) or *domain generalization* (Li et al., 2017), and is paramount to solving general intelligence.

In this paper, we focus on an instance of the domain generalization problem commonly met in video games: controlling an avatar and making it jump over an obstacle. Variations of this objective (or tasks) are obtained by changing the position of the obstacle. Humans will only need a few tries on a few tasks to estimate where the avatar needs to jump and will easily infer the strategy to follow when the obstacle is somewhere else on the screen. We show that the same cannot be said of a machine trained from scratch on the same few tasks with classic reinforcement learning techniques. Because of the limited number of tasks at hand, there are many ways to solve them while depending heavily on task-specific features and not on the generalizing ones that carry over between tasks. We call learning the features that truly matter for a given problem *learning invariances* and discuss alternatives to accomplish it. Our ultimate goal is the open problem of training an agent on a set of tasks as small as possible and having it generalize well to unseen obstacle positions.

In the following, we study the effectiveness of three methods to learn invariances for policy generalization: data augmentation, meta-learning and adversarial training. After a brief overview of related work, we describe the tasks we are trying to solve and the details of the methods we evaluated. Finally, we discuss their results on policy generalization.

## 2 RELATED WORK

**Zero-shot learning** is a topic of great interest these days. Among others, Oh et al. (2017) manually enforce analogies (resp. dissimilarities) between similar (resp. different) variables to reach better generalization on an 3-D labyrinth task. Higgins et al. (2017) apply $\beta$-VAE to extract independent latent variables from a scene and apply q-learning on those features. It is worth noting that independent features, while undoubtedly useful, are not sufficient on their own to guarantee generalization

as shall be seen on our toy problem. Additionally, it is notoriously difficult to apply q-learning to a small set of features, e.g. the RAM of an Atari game[1] (Sygnowski & Michalewski, 2016).

**Data augmentation** is a very common technique to induce invariances in supervised learning. On image recognition tasks, it usually involves randomly cropping and rotating the images of the data set, and results in agents with better generalization capabilities (Perez & Wang, 2017). In reinforcement learning, data augmentation has scarcely been applied. Silver et al. (2017) use the symmetries of the Go board to generate more games and arguably ensure the encoding of those symmetries in AlphaGo's policy and evaluation networks. Duan et al. (2017) pretrain a visual module on simulated objects with a variety of color, backgrounds, and textures in order to make the learnt representation independent from those attributes. That training is however done prior to the RL phase.

**Meta-learning** has been the focus of much attention recently. Finn et al. (2017) apply it to find model parameters such that a small number of gradient steps on a new task will produce good generalization performance on it. Li et al. (2017) adapt the method to zero-shot learning via a procedure aimed at aligning the gradient updates for two arbitrary training tasks and show some generalization improvement on Cart-Pole and Mountain Car. Compared to data augmentation which requires transforming images in somewhat sophisticated manners, the only supervision meta-learning requires is a task identifier, making it a broader technique.

In the wake of Goodfellow et al. (2014), much research has been directed towards **adversarial training**. The original idea is to simultaneously train a generator that captures the data distribution, and a discriminator that estimates whether a sample is real or generated. Ganin & Lempitsky (2015) apply adversarial training to domain adaptation and produce shift-invariant classifiers. Lample et al. (2017) extend the discriminator's duty to separate attributes (e.g. age, gender or glasses) from salient information in the latent space.

## 3 EXPERIMENT AND METHODS

**Setup.** We consider an extremely simple video game consisting of a black background, a floor and two rectangles (Fig.1 *Left*). The grey rectangle starts on the left of the screen and can be moved with two actions, "Right" and "Jump". The goal of this game is to reach the right of the screen while avoiding the white obstacle. There is only one specific distance (measured in number of pixels) to the obstacle where the agent has to chose the action "Jump" in order to pass over the obstacle. If jumping is chosen at any other point, the agent will inevitably crash into the obstacle. A reward of +1 is granted anytime the agent moves one pixel to the right (even in the air). The episode terminates if the agent reaches the right of the screen or touches the obstacle. We build a set of related tasks by varying two factors: the floor height and the position of the obstacle on the floor. The resulting set contains 1271 tasks. We use 6 of those for training and evaluate the generalization performance as the fraction of the remaining 1265 tasks the agent can solve.

**Explicit Invariance Learning Through State Augmentation.** We first attempt to guarantee generalization through data augmentation: we embed the original game screen at a random position in a larger black screen, and use that as input to classic RL algorithms. The larger screen has the shape of the game screen with $e$ pixels added in both dimensions (see Fig.1 *Left* for details). The position of the game screen is kept constant during each episode.

**Meta-learning Approach.** An interesting approach to zero-shot learning is to make sure that updates performed on one task are beneficial for solving other related tasks (Finn et al., 2017). In our case, one can hope that if getting better on a given position makes the agent better on another one, its overall ability to jump over arbitrary obstacles will also improve. In the spirit of Li et al. (2017), this translates into the following optimization:

$$\min_\theta L_1(\theta) + L_2(\theta - \alpha \nabla_\theta L_1) \tag{1}$$

where $\theta$ represents the weights of the neural network, $L_1$ and $L_2$ the losses on two random tasks (in our case, two positions of the obstacle), and $\alpha$ the learning rate. $\theta - \alpha \nabla_\theta L_1$ represents the weights after an update made to improve on task 1, we want that update to also decrease the loss on task 2. A first order approximation of this minimization reduces it to aligning the gradient updates (in

---

[1]A difficulty we also encountered during certain experiments.

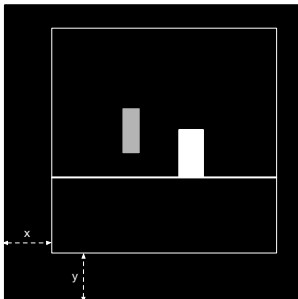 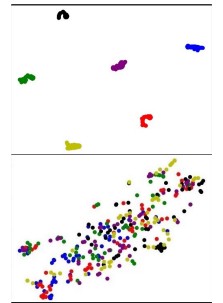 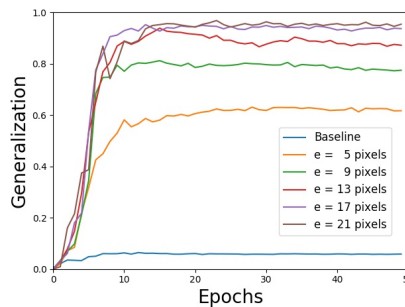

Figure 1: *Left.* Embedded game screen. The grey rectangle is the controllable object, the white one the obstacle. When data augmentation is performed, the game screen is embedded into a larger screen at a random position $(x, y)$ with $1 \leq x, y \leq e$. *Centre.* t-SNE plots of hidden layer outputs over the course of an episode with (bottom) and without (top) adversarial training. Colors correspond to training tasks. *Right.* Generalization performance of data augmentation for different values of $e$.

parameter space) obtained on different tasks, guiding the optimization to a more *generalizing* part of the function space (Li et al., 2017).

**Task Adversarial Learning.** Another interesting approach to this problem is to adversarially remove any task-specific information from the deep layers of the network. This is done through the addition of a discriminator trained to recognize the task instance from the output of a certain hidden layer. The RL agent then has two goals: solve the tasks and maximize the entropy of the discriminator's output. This approach is an RL equivalent to Lample et al. (2017)'s Fader Networks.

## 4 RESULTS AND DISCUSSION

We trained an agent[2] using standard Double-DQN (van Hasselt et al., 2015) and A2C (Mnih et al., 2016) with and without data augmentation for a variety of $e$ values. The generalization performance of the agent is shown on Fig.1 *Right.* Without data augmentation (blue curve) the agent solves all the training tasks, but only 36 out of the 1265 (2.8%) testing tasks. With **data augmentation**, it is able to solve up to 1263 of the 1265 variations (99.8%), confirming in a reinforcement learning setting that it is very helpful for generalization. Both DDQN and A2C produce the same patterns. Interestingly, the agent has also become able to jump over two consecutive obstacles, a situation never observed during training. **Meta-learning** and **adversarial training** on the other hand have so far failed to perform better than the baseline.

The one essential feature to learn in order to solve the jumping problem and generalize properly to any task is the *relative distance* between the controllable object and the obstacle. In other words, the agent needs to learn translation invariance. Other features are irrelevant and using them might actually prevent generalization. For instance, the object coordinates on the screen are sufficient to solve the problem in a variety of ways. One is to compute the difference between the two x-coordinates and learn which value should trigger a jump. Another is to discriminate between tasks using the coordinates of the obstacle, and then memorize for each task the absolute position where the object needs to jump. The former will generalize while the latter will not[3]. Our results show that through **data augmentation**, the agent becomes able to learn translation invariance. That invariance however has to be enforced manually, effectively trading supervision for generalization. Contrary to data augmentation, **meta-learning** is an invariance-agnostic training procedure and as such bears more promise. Additional work is required to understand the dynamics of learning with the meta-learning term and hopefully get better generalization. With **adversarial training**, the agent is successful at performing the training tasks and at masking their specific information: a t-SNE plot of hidden layer outputs shows clear tasks' clusters without adversarial training but none with (Fig.1 *Centre*). It appears though that the restriction provided by the adversary is not sufficient to generalize. Implicitly learning the proper invariance for generalization is still an open problem.

---

[2]The results shown are for the DQN network (Mnih et al., 2015) but other architectures perform equivalently.
[3]Learning independent features is thus in itself not enough to guarantee generalization (Higgins et al., 2017).

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
