# OpenReview forum: "Learning Invariances for Policy Generalization"
_ICLR.cc/2018/Workshop — Accept_

### Official Review · AnonReviewer3 · 2018-03-06
**Case study on policy generalization**

**Rating:** 7
**Confidence:** 3

**Review:**

The problem is chosen to be controlling an avatar and making it jump over an obstacle. The authors compare data augmentation, meta learning and adversarial training.
Pros
* Clear presentation and up to date references
* Interesting and difficult question
* promising results
Cons
* The chosen data augmentation method generates situations very similar to the testing problems which somehow biased the results
* The paper does not give hints for improving meta learning and adversarial training

---

### Official Review · AnonReviewer1 · 2018-03-08
**Nice experimental paper**

**Rating:** 7
**Confidence:** 4

**Review:**

The paper studies the problem of learning a RL agent which policy is invariant to certain transformation of the task and thus is potentially able to generalize to new variants of the task.
Authors introduce a simple task where a number of parameters can vary and conduct generalization experiments with double Q-learning agent as a model. The results show that the simplest data augmentation techniques improve generalization better than more complicated and principled approaches such as meta- or adversarial learning.
The core problem that is being studied seems very important to me and it is interesting that even such a simple "jumping" task presents a challenge. However, to gain more insights for the research community from the negative results being presented, the experiments should certainly contain more information. I understand that the short paper format maybe does not allow this fully, but maybe authors could use the appendix to specify the network architecture, the exact way they used more advanced techniques, how the double DQN agent has been trained exactly etc. This way we could know more if the negative result has some fundamental reasons or can potentially be fixed with more careful training regime.

---

### Decision · Program_Chairs · 2018-03-20
**ICLR 2018 Workshop Acceptance Decision**

**Decision:**

Accept

**Comment:**

Congratulations, your paper was accepted to the ICLR workshop.